# A Study on the Influence of Social Leisure Activities on the Progression to the Stage of Frailty in Korean Seniors

**DOI:** 10.3390/ijerph17238909

**Published:** 2020-11-30

**Authors:** AlChan Kim, Eunsurk Yi, Jiyoun Kim, MunHee Kim

**Affiliations:** 1Division of Sports Science, Baekseok University, Cheonan 31065, Korea; younal@empas.com; 2Department of Exercise Rehabilitation & Welfare, Gachon University, Incheon 21936, Korea; yies@gachon.ac.kr; 3Department of Health Science, Korea National Sport University, Seoul 05541, Korea

**Keywords:** frailty, aging, older adults, social leisure activities, national survey, Korean senior

## Abstract

In this study, we performed a logistic regression analysis according to the frequency of participation in social leisure activities (education, clubs, social groups, volunteer activities, religious activities, and senior citizens’ welfare center use) by men and women aged ≥ 65 years. We investigated the frequency of participation in social leisure activities and their association with the level of frailty (health vs. pre-frailty, health vs. frailty, pre-frailty vs. frailty). This study included 10,297 older adults (men: 4128, women: 6169) who participated in the 2017 National Survey of Older Koreans, and were divided into three groups (healthy, pre-frailty, and frailty). Five frailty index components were used to measure the frailty level. There was a positive relationship between the elderly’s religious activities, four times a week, from the healthy stage to the frailty stage, from the healthy stage to the pre-frailty stage, and from the pre-frailty stage to the frailty. In addition, positive associations emerged in leisure activities and club activities, respectively, from the healthy stage to the frailty stage (once a week, respectively). Positive association also emerged from the healthy stage to the pre-frailty and from the pre-frailty stage to the frailty stage (once a month to once in a two-week period).

## 1. Introduction

In recent years, with a rapidly aging society, the increasing number of older adults with deteriorating physical and psychological functions has emerged as a serious social problem. Aging literally means getting older. However, frailty refers to a condition in which the functions deteriorate severely, to the extent that it interferes with daily life [1]. Frailty is theoretically defined as a clinically recognizable state of increased vulnerability resulting from aging-associated decline in reserve and function across multiple physiologic systems, such that the ability to cope with every day acute stressors is compromised. In the absence of a gold standard, frailty has been operationally defined by Fried et al. as meeting three out of five phenotypic criteria, indicating compromised energetics: low grip strength, low energy, slowed waking speed, low physical activity, and/or unintentional weight loss [2].

In the United States, 7% of the older adults aged 65 years or older become frail, and this increases with age, indicating that one in four (25%) older adults aged 85 years or older are frail [3,4]. The prevalence of frailty among those aged 65 years and above in Korea is 8.3%, and the risk of frailty increases with age. Although the frequency of frailty among Korean senior citizens varies depending on the age measurement scale, and is measured using the five components of the Frailty Index (fatigue, resistance, ambulation, illness, loss of weight), it ranges from 21.9% to 29.0% [5].

The main symptoms associated with frailty are decreased muscle mass, weight loss, muscle weakness, fatigue, decrease in walking speed, and decrease in physical activity. Frailty has a high risk of adverse health consequences because it degrades several homeostasis functions, making them vulnerable to stressors [2,6]. The pandemic of infectious diseases, such as the coronavirus disease 2019 (COVID-19) causes enormous damage in older adults in terms of mortality. Particularly in the elderly with the disease, frailty has serious clinical symptoms, making death from COVID-19 even more likely [7].

Frailty can lead to disease, hospitalization, and activity restrictions, leading to gradual decline in functions [8,9]. Risk factors for frailty include age, sex, disease, social factors (such as loneliness, social isolation), economic factors, malnutrition, low levels of physical activity, and long sedentary lifestyles [10,11,12].

Not only does frailty weaken the functioning of various body organs, but it also results in the lack of ability to adapt to leisure and social activities, making it impossible for the older adults to live healthy in daily life. Frailty is divided into three stages: healthy, pre-frailty, and frailty. In order to slow down the progression from the healthy or pre-frailty stages in older adults to the next stage, frailty-related factors should be managed through various activities in the leisure time of older adults [13]. After retirement, older adults are often confused because of the loss of their official role, changes in family relations, or loss of their spouse, and may face the problem of how to spend a satisfactory retirement period [14]. In particular, the network of relationships, which is an important support base for old age adaptation, will be slowed, with emotional ties becoming estranged. These psychological and social stresses eventually lead to the loss of physical function, leading to chronic diseases or lower satisfaction levels in life [15].

For successful aging, it is importance for older adults to participate in active activities to prevent frailty and maintain physical independence. In addition, to prevent aging, it is necessary to know and manage aging predictors in advance. Social factors, such as social participation, social support, and physical fitness management have been reported as predictors of frailty [16,17,18,19].

Therefore, the study analyzed the frequency of participation in social leisure activities (education, clubs, social groups, volunteer activities, religious activities, use of senior citizen centers/senior citizen welfare centers) by frailty stage in Koreans aged over 65 years who participated in the 2017 national survey. The purpose of this study was to determine the relationship between appropriate frequency of participation in social leisure activities

## 2. Materials and Methods

### 2.1. Participants

This study included 10,297 older men and women aged 65 years or older based on the results of the 2017 National Survey of Older Koreans conducted by the Ministry of Health and Welfare of Korea (Korea Institute for Health and Social Affairs; 12, Seobukbu 2-ro, Jochiwon-eup, Sejong-si, Republic of Korea). The study participants were divided into groups, according to the three stages of frailty based on the five components of the Component Frailty Index [3].

The three groups that the participants were divided into based on the frailty stages were: the healthy, pre-frailty, and frailty stages with 735 (7.1%), 7746 (75.2%), and 1816 (17.6%) participants, respectively. As shown in Table 1, among the 2017 National Survey of Older Korean respondents, 40.1% (*n* = 4128) were men and 59.9% (*n* = 6169) were women. The participants’ average age was 74.6 ± 9.8 years, most participants were married (62.2%, *n* = 6405), and had graduated from elementary school (34.9%, *n* = 3597).

### 2.2. Study Design and Process

#### 2.2.1. Sampling and Survey Method

This study, based on the 2017 National Survey of Older Koreans, targeted older adults aged over 65 years. The sample was set using the stratified cluster sampling, which is a method of stratification using primary extraction units, which are extracted from each layer as a colony. The survey period was from June to August 2017. Investigators visited the selected households in person. For those who could not respond directly, the nearest household member was asked to respond by proxy. However, a proxy response was not allowed for such survey items as cognitive function and depression that require subjective judgment. Data of 10,297 older Korean individuals were used (excluding two for whom the five FI components could not be applied, out of the 10,299 surveyed).

This study used only data of leisure and social activities in the questionnaires. In survey data questions on leisure and social activities, frequency of participation by education program, clubs, social groups, volunteer activities, religious activities, and the use of senior citizens’ welfare centers were determined.

#### 2.2.2. Frailty Index

The three factors for determining the reference setting for the measurement index that was proposed for frailty measurement were: weight loss of ≥4.54 kg reduction; walking speed ≥ (9 s); and grip strength (average ± standard deviation [SD] minimum 20%) kg; sex; and body mass index). The five FI components analyzed in this study included judgment elements, and Table 2 shows the definition of the measurement index and the setting of the standards [3].

The 5 FI components for the frailty scale measurement variables include fatigue (fatigue exhaustion), ambulation walking speed (4 m), balance or resistance (resistance timed up and go (TUG) test score > 10 s), disease and drug use (illness poly-pharmacy), and weight. One point was assigned, if applicable, to the relevant survey item among the 5 items; or 0 point if not. The relevant survey question numbers for the five items are described in Table 2. One point was assigned if there was a relevant item among the five items, and 0 otherwise. A total score of the five items of 0 refers to a healthy stage; between 1 and 2 is the pre-frailty stage; and 3 or more points refer to the frailty stage.

#### 2.2.3. Social Leisure Activities

The factors constituting social leisure activities in this study are education, clubs, social groups, polytechnic and social organizations, volunteering activities, religious activities, and senior citizen centers/senior wellness centers. The definitions for the subsets are shown in Table 3.

### 2.3. Ethics Statement

To conduct this study, we downloaded the raw 2017 National Survey of Older Korean data, collected by the Ministry of Health and Welfare of Korea (https://www.kihasa.re.kr/r/). To use the data, we followed the procedure for using the original data on the Institute for Health and Social Affairs website, from June to August 2017. The survey on the status of the elderly, a state-approved statistic, is a court survey, performed in accordance with Article 5 of the Elderly Welfare Act, and conducted every three years after legalization in 2007.

### 2.4. Statistical Analysis

In this study, among the independent variables, age was categorized as youngest-old (65–74), middle-old (75–84), and oldest-old (85 years or older). The remaining independent variables (education, clubs, social groups, polytechnic and social organizations, volunteering activities, religious activities, senior citizen centers/senior wellness centers) were included from the questionnaire in the 2017 National Survey of Older Koreans. The general characteristics of the study participants were calculated through descriptive statistics and frequency analysis. The value of the frailty measurement reference item was recoded as “0” for “Yes” and “1” for “No.” Binary logistic regression analysis was performed to calculate the risk of independent variables that affect frailty stages (healthy, pre-frailty, and frailty stages) with the recoded values. The comparison between each stage of frailty was conducted for “healthy vs. pre-frailty stage,” “healthy vs. frailty stage,” and “pre-frailty and frailty stage.” All statistical analyses were performed using the IBM SPSS Statistical for Windows, version 23.0 (IBM Corp., Armonk, NY, USA), and the significance level was set at *p* < 0.05.

## 3. Results

This study was performed to determine the relationship between appropriate frequency of participation of social leisure activities. Logistic regression analysis was performed three times (healthy vs. pre-frailty, healthy vs. frailty, and pre-frailty vs. frailty).

Women (rather than men) and older age participants were more likely to be associated with frailty. Among the social leisure activities, when the frequencies of participation in clubs, social groups, and religious activities were once a month, once every two weeks, and more than four times a week, respectively, the possibility of progressing into the frailty stage was the slowest.

Table 4 presents the results of the logistic regression analysis according to the frequency of use; with education program, clubs, social groups, volunteer activities, religious activities, and senior citizens’ welfare centers as predictors of frailty stage compared to the healthy stage. Women were about 2.9 times more likely to be frail than men. The probability of frailty compared to the healthy stage was low when club participation occurred once a month, when social group participation occurred once every two weeks, and religious activities occurred four or more times a week.

Table 5 presents the results of the logistic regression analysis including sex, age, clubs, social groups, participation in religious activities, and frequency of use per senior citizen as predictors of pre-frailty stage compared to the healthy stage. Women were about 1.9 times more likely to be frail than men. The probability of pre-frailty compared to the healthy stage was low if club participation occurred once fortnightly, social group participation occurred once a week, and religious activities occurred four or more times a week.

Table 6 presents the results of the logistic regression analysis including sex, age, frequency of use per senior citizen, club, social group, and frequency of religious activities as predictors of frailty stage compared to the pre-frailty stage. Women were about 1.4 times more likely to be frail than men. The probability of frailty compared to the pre-frailty stage was low if club participation occurred once a month, social group participation occurred once fortnightly, and religious activities occurred four or more times a week.

Women, rather than men, and those in the older age groups were found to be more likely to progress into the frailty stage from the pre-frailty stage. Among the factors of social leisure activities, of those who progressed into the frailty stage from the pre-frailty stage compared to the healthy, the rate was the slowest for those who participated six times a week. When the frequencies of participation in clubs, social groups, and religious activities were once a month, once every two weeks, and more than four times a week, respectively, the possibility of progressing to the frailty stage from the pre-frailty stage was the slowest.

## 4. Discussion

The increase in the population of frail individuals due to a rapidly aging society is becoming serious social problems, including increase in medical expenses. This study predicted frailty according to the frequency of social leisure activities (education, clubs, social groups, volunteer activities, religious activities, and senior citizens’ welfare centers) by age group, for men and women aged 65 years or older. The purpose of this study was to determine the relationship between the frequency of appropriate participation in social leisure activities required to slow the progression from the healthy stage to the pre-frailty stage and the stage of frailty.

The main purpose of leisure activity is to build relationships, including socializing with peers and strengthening human relationships [3]; therefore, enriching the opportunities for such interactions is very important for the psychological well-being.

In particular, social participation and social relations become important factors in improving the life satisfaction of older adults whose feelings of alienation and loneliness are problems, arising due to the disconnection of relationships and loss of roles as aging progresses [20]. Therefore, it is important to improve the life satisfaction by enhancing the quality of social relations through leisure [21].

Social activities, such as leisure life, volunteer work, employment, and religion are proposed for effective retirement. Of these, religious activities are the most preferred by older adults, and religious institutions are sources of emotional support for them, after family and the state [22]. This study also found that participation in religious activities four times a week was the best way to slow the progression from the healthy stage to the pre-frailty or frailty stages.

The religious participation rate includes about half (53.1%) of the entire Korean population, 63.3% of whom are older adults aged over 60 years; thus, religion has an important meaning in retirement [23]. Securing a social network that can support older individuals is very important, and participation of older adults in religious activities often provide them with the opportunity to feel a sense of value and efficacy for themselves while experiencing interactions with others. For the elderly in Korean society, religion is not only the beginning of social relationships, but also has a profound relationship with the continuation of emotional relationships [24].

These religious activities reinforce subjective well-being [25], have a positive effect on the quality of life [26], reduce depression and psychological stress [27], and decrease the mortality rate of older adults. Religious activities also seem to be helpful for the maintenance of physical and mental health and the quality of life of older adults [28].

Leisure time enjoyed together in social relationships extends beyond limited relationships in clubs or social groups to friends and community relations; as well as social activities conducted while interacting with other people, which lead to life satisfaction for older individuals [29].

In this study, in order to slow the progression into the frailty stages from the healthy stage, participation in social activities during leisure time occurred once a week, whereas participation in club activities was once every two weeks, which were beneficial. To slow progression to the pre-frailty stage from the healthy stage, participation in social activities during leisure time occurred once every two weeks, whereas participation in club activities was once a month, which were beneficial.

Participation in club activities and social groups helps to enhance strength and vitality following participation in meaningful activities and hobbies or sharing concerns with one’s peers. It also helps to slow social and emotional isolation by restoring the dwindling social networks. The results of this study recommended the frequency of use of social activities per senior citizen at six or seven times per week. This is because older adults living in similar areas may have more opportunities to be involved with direct personal life exchanges or share minor concerns with each other, or cooperate with one another. The participation of the elderly in social leisure activities is thought to play a major role in getting the elderly to have a positive and active life attitude rather than a negative view of themselves.

The limitations of this study and some suggestions for subsequent studies are as follows. There was no information on the period of participation in social leisure activities of the elderly who participated in the survey. In future, it is important to consider such things as the duration of participation and detailed activities of the elderly. Studies are also needed to identify ways of reducing the risk of frailty by including data on various quantitative and qualitative variables such as health and behavior habits and physical activities, as well as social leisure activities that can affect frailty. In addition, since the average life expectancy of the elderly population is increasing, it is necessary to closely monitor the various patterns that are changing over time through a longitudinal follow-up survey rather than a cross-sectional study on frailty. In addition, it is necessary for the community and the country to develop continuous and new frailty prevention programs by investigating the various daily activities of vibrant 100-year-olds.

## 5. Conclusions

In this study, logistic regression analysis was performed for men and women aged 65 years or older according to their frequency of participation in social leisure activities (education, clubs, social groups, volunteer activities, religious activities, and use of senior citizens’ welfare centers). As a result of this study, in order to slow the relationship from the healthy stage to pre-frailty and frailty stages, older people commonly need to engage in religious activities four times a week. In order to slow the relationship from the healthy stage into the frailty stages, participation in social activities during leisure time should occur once a week, whereas participation in club activities should occur once every two weeks. From the healthy stage to the pre-frailty stages, participation in social activities during leisure time should occur once every two weeks, while participation in club activities should occur once a month. From the pre-frailty stage to the frailty stage, participation in social activities during leisure time should occur once every two weeks, whereas participation in club activities should be once a month. The participations in activities at the stated frequencies would be beneficial.

## Figures and Tables

**Table 1 ijerph-17-08909-t001:** General characteristics of the study participants.

Variable	Classification	% (*n*)
Sex	Male	40.1% (4128)
	Female	59.9% (6169)
Age (years)	74.6 (Mean)	±9.80 (SD)
Marital status	Single	0.5% (49)
	Married	62.2% (6405)
	Bereaved	33.4% (3439)
	Divorced	3.1% (315)
	Separated	0.9% (88)
	Others	0.0% (1)
Educational level	Not knowing alphabets	7.8% (805)
	Reading characters	19.4% (1995)
	Elementary school	34.9% (3597)
	Middle school	15.8% (1625)
	High school	15.7% (1612)
	College	1.0% (98)
	University	5.5% (565)

SD, standard deviation.

**Table 2 ijerph-17-08909-t002:** The definition and standards of the 5 FI (Frailty Index) components for the frailty scale.

Variable	Definition	Criteria	Survey Question Number
Fatigue exhaustion	Did you often feel tired during the recent month?	Frailty ≥ 3 items pre-frailty 1–2 items Healthy = 0	Fatigue B6-2
Ambulation walking speed (4 m)	Did you have any difficulty climbing up 10 steps by yourself and without using aids?		Resistance D5-3
Resistance TUG test score (>10 s)	1. Did you have any difficulty walking 200 or 300 m by yourself and without using aids? 2. Illness: “Did a doctor tell you that you have the following 11 illnesses?		Ambulation D5-2 B2-3
Illness poly-pharmacy	The illnesses included hypertension, diabetes, angina, heart attack, congestive heart failure, stroke, chronic lung disease, asthma, arthritis, cancer, kidney disease		Illness B2-1-1 B2-1-4 B2-1-5 B2-1-6 B2-2-5 B2-2-2 B2-1-11 B2-1-12 B2-1-8 B2-1-7 B2-1-21
1. Weight loss 2. Malnourished and risk of malnutrition	MNA-SF; Short Form Mini Nutritional Assessment		Loss of Weight C5-9 C5-1-10

TUG, timed up and go test.

**Table 3 ijerph-17-08909-t003:** Definition of social leisure activities.

Participating Item	Definition
Education program	All forms of organized educational activities other than school education
Clubs	Gathering of people who share the same hobby and enjoy it together
Social groups	Orderly group for the purpose of being intimate and harmonious with each other
Political and social organizations	Organization aimed at resolving political or social problems
Volunteering activities	Intentional and planned daily activities that are voluntary actions by free will and actively participate in the creation of a welfare society
Religious activities	All activities for religion
Senior citizen centers/senior welfare centers	The act of visiting a senior citizen center or welfare center for the elderly for the purpose

**Table 4 ijerph-17-08909-t004:** Logistic regression analysis according to health and frailty stages.

Factor			B	S.E.	Exp (B)	95% of EXP (B) C.I.	
						Lower	Upper
Sex	Men (*n* = 1004, 39.4%)				1		
	Women (*n* = 1547, 60.6%)		1.078	0.119	2.940 ***	2.327	3.714
Age	65~74 years (*n* = 1148, 45.0%)				1		
	75~84 years (*n* = 1122, 44.0%)		1.407	0.119	4.086 ***	3.234	5.162
	85 years or older (*n* = 281, 11.0%)		1.801	0.268	6.053 ***	3.577	10.244
Social Leisure Activity	Frequency of participation in education	None (*n* = 2266, 88.8%)			1		
		Less than once a month (*n* = 1, 0.0%)	22.696	40,192.969	7,189,447,742.02	0.000	.
		Once a month (*n* = 3, 0.1%)	22.305	18,939.906	4,863,317,491.69		.
		Once a fortnight (*n* = 6, 0.2%)	0.291	1.584	1.338	0.060	29.802
		Once a week (*n* = 109, 4.3%)	0.190	0.284	1.209	0.693	2.107
		2–3 times a week (*n* = 129, 5.1%)	0.328	0.249	1.388	0.853	2.260
		4 or more times a week (*n* = 37, 1.5%)	−0.640	0.437	0.527	0.224	1.242
	Frequency of club participation	None (*n* = 2455, 96.2%)			1		
		Less than once a month (*n* = 9, 0.4%)	−2.083	1.106	0.125	0.014	1.088
		Once a month (*n* = 29, 1.1%)	−2.487	0.790	0.083 *	0.018	0.391
		Once a fortnight (*n* = 9, 0.4%)	−20.561	13,049.056	0.000	0.000	.
		Once a week (*n* = 16, 0.6%)	−0.674	0.624	0.510	0.150	1.731
		2–3 times a week (*n* = 18, 0.7%)	−0.552	0.589	0.576	0.182	1.825
		4 or more times a week (*n* = 15, 0.6%)	−1.593	0.756	0.203 *	0.046	0.895
	Frequency of participation in social groups	None (*n* = 1712, 67.1%)			1		
		Less than once a month (*n* = 189, 7.4%)	−1.053	0.186	0.349 ***	0.242	0.502
		Once a month (*n* = 406, 15.9%)	−1.169	0.140	0.311 ***	0.236	0.409
		Once a fortnight (*n* = 133, 5.2%)	−1.461	0.214	0.232 ***	0.153	0.353
		Once a week (*n* = 72, 2.8%)	−1.434	0.285	0.238 ***	0.136	0.417
		2–3 times a week (*n* = 22, 0.9%)	−0.528	0.483	0.590	0.229	1.520
		4 or more times a week (*n* = 17, 0.7%)	−1.169	0.546	0.311 *	0.106	0.907
	Frequency of participation in religious activities	None (*n* = 2471, 97.3%)			1		
		Less than once a month (*n* = 10, 0.4%)	−0.058	0.156	0.944	0.695	1.281
		Once a month (*n* = 18, 0.7%)	−0.105	0.264	0.900	0.536	1.510
		Once a fortnight (*n* = 6, 0.2%)	−0.175	0.483	0.839	0.326	2.161
		Once a week (*n* = 13, 0.5%)	−0.329	0.154	0.720 *	0.532	0.974
		2–3 times a week (*n* = 16, 0.6%)	−0.462	0.210	0.630 *	0.418	0.951
		4 or more times a week (*n* = 7, 0.3%)	−0.830	0.295	0.436 **	0.245	0.778

* *p* < 0.05, ** *p* < 0.01, *** *p* < 0.001. S.E., standard error; CI, confidence interval.

**Table 5 ijerph-17-08909-t005:** Logistic regression analysis according to health and pre-frailty stages.

Factor			B	S.E.	Exp (B)	95% of EXP (B) C.I.	
						Lower	Upper
Sex	Men (*n* = 3562, 42.0%)				1		
	Women (*n* = 4919, 58.0%)		0.637	0.087	1.892 ***	1.594	2.244
Age	65~74 years (*n* = 4757, 56.1%)				1		
	75~84 years (*n* = 3251, 38.3%)		0.828	0.098	2.289 ***	1.888	2.776
	85 years or older (*n* = 473, 5.6%)		0.869	0.244	2.384 ***	1.477	3.850
Social Leisure Activity	Frequency of use senior community center	None (*n* = 6329, 74.6%)			1		
		1 time (*n* = 250, 2.9%)	0.309	0.259	1.363	0.820	2.265
		2 times (*n* = 242, 2.9%)	0.149	0.254	1.160	0.705	1.909
		3 times (*n* = 410, 4.8%)	0.434	0.231	1.543	0.980	2.429
		4 times (*n* = 191, 2.3%)	0.527	0.369	1.694	0.823	3.489
		5 times (*n* = 469, 5.5%)	0.812	0.269	2.253 **	1.329	3.820
		6 times (*n* = 140, 1.7%)	0.551	0.462	1.736	0.701	4.295
		7 times (*n* = 450, 5.3%)	0.717	0.263	2.048 **	1.224	3.426
	Frequency of club participation	None (*n* = 8089, 95.4%)			1		
		Less than once a month (*n* = 39, 0.5%)	−0.684	0.413	0.505	0.225	1.133
		Once a month (*n* = 130, 1.5%)	−0.0601	0.231	0.548 **	0.349	0.862
		Once a fortnight (*n* = 32, 0.4%)	−1.133	0.410	0.322 **	0.144	0.719
		Once a week (*n* = 75, 0.9%)	−0.285	0.339	0.752	0.387	1.460
		2–3 times a week (*n* = 62, 0.7%)	−0.572	0.344	0.564	0.287	1.108
		4 or more times a week (*n* = 54, 0.6%)	−0.928	0.346	0.395 **	0.201	0.779
	Frequency of participation in social groups	None (*n* = 4437, 52.3%)			1		
		Less than once a month (*n* = 862, 10.2%)	−0.266	0.131	0.766 *	0.593	0.991
		Once a month (*n* = 1975, 23.3%)	−0.330	0.100	0.719 ***	0.592	0.874
		Once a fortnight (*n* = 701, 8.3%)	−0.335	0.139	0.716 *	0.545	0.939
		Once a week (*n* = 342, 4.0%)	−0.394	0.180	0.674 *	0.474	0.960
		2–3 times a week (*n* = 102, 1.2%)	−0.220	0.335	0.802	0.416	1.546
		4 or more times a week (*n* = 62, 0.7%)	−0.434	0.370	0.648	0.314	1.337
	Frequency of participation in religious activities	None (n = 3894, 45.9%)			1		
		Less than once a month (*n* = 1676, 19.8%)	0.077	0.112	1.081	0.868	1.345
		Once a month (*n* = 365, 4.3%)	−0.0315	0.187	0.730	0.506	1.053
		Once a fortnight (*n* = 116, 1.4%)	−0.034	0.331	0.966	0.505	1.849
		Once a week (*n* = 1365, 16.1%)	−0.251	0.112	0.778 *	0.625	0.969
		2–3 times a week (*n* = 722, 8.5%)	−0.052	0.157	0.950	0.698	1.291
		4 or more times a week (*n* = 343, 4.0%)	−0.374	0.189	0.688 *	0.475	0.997

* *p* < 0.05, ** *p* < 0.01, *** *p* < 0.001. S.E., standard error; CI, confidence interval.

**Table 6 ijerph-17-08909-t006:** Logistic regression analysis according to pre-frailty and frailty stages.

Factor			B	S.E.	Exp (B)	95% of EXP (B) C.I.	
						Lower	Upper
Sex	Men (*n* = 3690, 38.6%)				1		
	Women (*n* = 5872, 61.4%)		0.327	0.061	1.386 ***	1.230	1.563
Age	65~74 years (*n* = 4779, 50.0%)				1		
	75~84 years (*n* = 4067, 42.5%)		0.627	0.061	1.872 ***	1.660	2.111
	85 years or older (*n* = 716, 7.5%)		0.973	0.096	2.647 ***	2.195	3.193
Social Leisure Activity	Frequency of use of senior community center	None (*n* = 7092, 74.2%)			1		
		1 time (*n* = 270, 2.8%)	−0.473	0.186	0.623 *	0.432	0.898
		2 times (*n* = 269, 2.8%)	−0.329	0.174	0.720	0.512	1.011
		3 times (*n* = 475, 5.0%)	−0.360	0.129	0.698 **	0.542	0.899
		4 times (*n* = 218, 2.3%)	−0.564	0.193	0.569 **	0.389	0.831
		5 times (*n* = 552, 5.8%)	−0.477	0.121	0.621 ***	0.489	0.787
		6 times (*n* = 159, 1.7%)	−0.704	0.231	0.494 **	0.314	0.778
		7 times (*n* = 527, 5.5%)	−0.490	0.124	0.612 ***	0.480	0.780
	Frequency of club participation	None (*n* = 9228, 96.5%)			1		
		Less than once a month (*n* = 32, 0.3%)	−1.370	1.026	0.254	0.034	1.898
		Once a month (*n* = 107, 1.1%)	−1.314	0.592	0.269 *	0.084	0.858
		Once a fortnight (*n* = 23, 0.2%)	−19.187	8139.473	0.000	0.000	.
		Once a week (*n* = 69, 0.7%)	−0.451	0.476	0.637	0.251	1.617
		2–3 times a week (*n* = 58, 0.6%)	0.002	0.417	1.002	0.443	2.268
		4 or more times a week (*n* = 45, 0.5%)	−0.511	0.610	0.600	0.181	1.983
	Frequency of participation in social groups	None (*n* = 5603, 58.6%)			1		
		Less than once a month (*n* = 859, 9.0%)	−0.654	0.118	0.520 ***	0.412	0.655
		Once a month (*n* = 1941, 20.3%)	−0.799	0.087	0.450 ***	0.379	0.533
		Once a fortnight (*n* = 672, 7.0%)	−0.995	0.152	0.370 ***	0.275	0.498
		Once a week (*n* = 326, 3.4%)	−0.777	0.206	0.460 ***	0.307	0.689
		2–3 times a week (*n* = 102, 1.1%)	−0.561	0.327	0.571	0.301	1.084
		4 or more times a week (*n* = 59, 0.6%)	−0.398	0.415	0.672	0.298	1.516
	Frequency of participation in religious activities	None (*n* = 4539, 47.5%)			1		
		Less than once a month (*n* = 1855, 19.4%)	−0.158	0.077	0.854 *	0.735	0.992
		Once a month (*n* = 400, 4.2%)	0.044	0.142	1.045	0.790	1.381
		Once a fortnight (*n* = 124, 1.3%)	−0.205	0.266	0.815	0.484	1.372
		Once a week (*n* = 1512, 15.8%)	−0.146	0.080	0.864	0.739	1.010
		2–3 times a week (*n* = 787, 8.2%)	−0.286	0.111	0.751 **	0.604	0.934
		4 or more times a week (*n* = 345, 3.6%)	−0.588	0.180	0.555 ***	0.390	0.791
	Frequency of participation in volunteer work	None (*n* = 9232, 96.5%)			1		
		Less than once a month (*n* = 54, 0.6%)	−0.312	0.536	0.732	0.256	2.093
		Once a month (*n* = 72, 0.8%)	−0.495	0.523	0.610	0.219	1.698
		Once a fortnight (*n* = 29, 0.3%)	−0.266	0.754	0.767	0.175	3.360
		Once a week (*n* = 77, 0.8%)	−1.154	0.598	0.315	0.098	1.019
		2–3 times a week (*n* = 63, 0.7%)	0.369	0.380	1.446	0.686	3.045
		4 or more times a week (*n* = 35, 0.4%)	−1.551	1.027	0.212	0.028	1.586

* *p* < 0.05, ** *p* < 0.01, *** *p* < 0.001. S.E., standard error; CI, confidence interval.

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
