# Peer review of "A Study on the Influence of Social Leisure Activities on the Progression to the Stage of Frailty in Korean Seniors"

_ijerph, 2020, doi:10.3390/ijerph17238909_

Round 1

Reviewer 1 Report

The subject is of interest great in the literature; however, the methods are not well described and the statistical analysis is not adequate. I suggest the authors review the paper and resubmit.

In the Methods section:

  • Describe how the subjects were classified according to the frailty status and how the data for this classification was collected. It is not clear in the paper. Include references.
  • How were the independent variables collected and how they were categorized? (What were the questions and the possible answers?) Review the categorization of the independent variables.
  • Multinomial regression analyses, including the three levels of frailty status, should be used to explore the associations between social participation and frailty status. The variables age, sex, marital status, and educational level should be used as adjustment variables. They are cofounders.
  • The interpretation of the results should be in line with the design of the study – cross-sectional and not longitudinal.

PINTO, Juliana Martins and NERI, Anita Liberalesso. Factors related to low social participation in older adults: findings from the Fibra study, Brazil. Cad. saúde colet. [online]. 2017, vol.25, n.3 [cited 2020-10-24], pp.286-293.

Marsh C, Agius PA, Jayakody G, Shajehan R, Abeywickrema C, Durrant K, Luchters S, Holmes W. Factors associated with social participation amongst elders in rural Sri Lanka: a cross-sectional mixed-methods analysis. BMC Public Health. 2018 May 16;18(1):636. DOI: 10.1186/s12889-018-5482-x. PMID: 29769054; PMCID: PMC5956789.

Reviewer 2 Report

  1. The authors should consider their definition of frailty and cite literature that posits frailty as a state of increased vulnerability. The authors are using a frailty definition from 1974 and there has been significant process made in the last 36 years. 
  2. It is unclear how and why COVID would factor into this study and I'd suggest omitting it from the introduction. 
  3. The study aims are written as a causal statement which can't be investigated using a secondary data analysis. I would encourage the authors to re-write the aims to explore association vs. causality. 
  4. The educational level of not knowing the alphabet vs. reading characters does not seem to align with formal definitions of literacy. Is there a way to formalize this or provide more information, esp. for non Korean readers? 
  5. There is no discussion of what the difference between participation in education, club participation, social groups. 
  6. What is the difference between once a month and less than once a month (is less than once a month meaning a few hours or half a day?) 
  7. The authors appear to have separated out non-frail to pre frail and then pre-frail to frail. It would be helpful to have a comparison across all 3 groups. 
  8. The authors should also enhance the background to talk about why leisure activities would matter for reduce frailty and what the proposed mechanism of change would be. The discussion addresses this well, but there is little information in the background on previous work that explores the relationship between leisure time activities and frailty. 
  9. The conclusion points are worded in strongly causal language and the authors may wish to temper their wording to suggest certain activities. 
  10. The importance of this paper does not come through as clearly as it potentially could. For years, we've talked about the importance of exercise and physical activity to reduce frailty. The results from this paper suggest that social activities may be a therapeutic target for frailty interventions. This could be discussed more in the results. 

Round 2

Reviewer 1 Report

The paper is of great interest, but there was no significant improvement from the last version.

The description of how the participants were categorized in healthy, pre-frailty, and frailty is still not clear. Did the authors use more than one criterion?

The description of the results should be reviewed.

Other issues are pointed out in the manuscript.

Author Response

Thank you for your faithful review.

To explain the classification of the group, the survey question number was entered in table2 and the description was added accordingly. The research results were supplemented as well. And the modified part was highlighted.

best regards

Reviewer 2 Report

The authors have addressed all comments provided by the reviewers. 

Author Response

Thank you for your faithful review. Additionally, the revised and supplemented content is highlighted.

Best regards
